# Elite sport hubs during COVID-19: The job demands and resources that exist for athletes

**Samantha Marshall** [1]*, **Nicola McNeil**[2], **Emma Louise Seal** [3], **Matthew Nicholson**[4]

**1** Centre for Sport and Social Impact, La Trobe University, Melbourne, Victoria, Australia, **2** Department of Management and Marketing, La Trobe University, Melbourne, Victoria, Australia, **3** Social and Global Studies Centre, RMIT University, Melbourne, Victoria, Australia, **4** Malaysia Office of the PVC & President, Monash University Malaysia, Subang Jaya, Selangor, Malaysia

* Samantha.Marshall@latrobe.edu.au

**Data Availability Statement:** All relevant data are within the paper.

**Funding:** This study was supported by a PhD Industry Scholarship with Basketball Australia. The funders had no role in study design, data collection

## Abstract

In response to the COVID-19 pandemic, elite sport leagues implemented hubs, or 'bubbles', which restricted athletes' movements and social interactions in order to minimise the risk of athlete infection and allow competitions to continue. This was a new way of working and living for elite athletes and there was a dearth of literature on this topic. The main objective of the study was to investigate the impacts of the hub model on athletes over time, and what job demands and resources existed for athletes through the application of Demerouti et al. (2001) Job Demands-Resources Model. Multiple sequential semi-structured interviews were conducted with Women's National Basketball League (WNBL) athletes during the 2020 season, which was held entirely in a hub in North Queensland, Australia. The key job demands in an elite sport hub identified were the volume of work, simultaneous overload and underload, and nature of work in the hub. The key resources that emerged include recovery services, control and player agency, and constructive social relations. Despite the presence of job resources, which work to counteract, or buffer job demands in order to reduce work stress and improve motivation, they were found to be insufficient for athletes and inequitably distributed between clubs. The intensity of the hub model also amplified demands present in all WNBL seasons. This research is therefore useful for planning of future elite sport leagues to improve the type and amount of resources available to athletes, thereby improving athlete wellbeing and performance both within and outside a hub model.

## Introduction

In response to the coronavirus (COVID-19) pandemic, elite sporting competitions had to adapt quickly to survive. During the initial stages of the pandemic, sport was cancelled or postponed in many cases, from grass roots community competitions to major international events such as the 2020 Tokyo Olympics [1–4]. Many athletes rely on competitions and ensuing performance bonuses to survive financially, and the cancellation of large sporting events has had significant social and economic impacts on clubs and associations at all levels [1, 3]. Indeed, the pandemic has been conceptualized as a 'longitudinal, multifaceted, unpredicted, non-

and analysis, decision to publish, or preparation of
the manuscript.

**Competing interests:** The authors have declared
that no competing interests exist.

controlled change event' which impacted athlete careers and had varying impacts on the mental health of athletes [5, 6]. The pandemic is likely to have had an emotional and psychological impact on many elite athletes, who were isolated from their teammates and social networks, had lower quality interactions with coaches, and risked potential infection [7]. When sport resumed, many leagues implemented hubs, or 'bubbles', to minimize risk of infection between athletes and the community, but little is known about what effect this had on athletes. The purpose of this study is to understand how the experience of the 'hub' environment impacts on elite female athletes during a condensed, national competition. More specifically, we explore the nature of the job demands placed on these athletes, the resources available to them to meet these demands, and the resultant impact on their wellbeing, through the application of the Job Demands and Resources model [8, 9].

## Impact of COVID-19 on sport

The Covid-19 pandemic impacted athletes' training and competition, affecting their fitness and skill maintenance and development, as well as their mental health [10]. In sport, as in the wider community, the pandemic has and continues to have a more severe impact on women, financially and in terms of performance and support structures [4, 11, 12]. In their study of 95 elite UK sportswomen during the pandemic, Bowes et al. [13] found that 65% experienced a loss of income, which was significant because of their already precarious financial situation. This financial hardship was compounded by a lack of access to equipment when training from home—women athletes often had to buy their own equipment, or use sub-optimal equipment to train remotely, whereas comparable men's teams were provided with equipment by their club or union [13]. Psychologically the pandemic increased demands on athletes, both elite and amateur, and women athletes were found to be more stressed, and distressed, than male athletes [14]. Additionally, women's sport is often perceived as being less commercialized, or less profitable, than men's sports, and as such, less urgency was placed on the continuation of women's leagues at the start of the pandemic [15, 16]. For example, the Australian Football League Women's (AFLW) competition was cancelled without completing the finals series after their nine-week regular season. However, the men's league was only temporarily suspended and later resumed its 18-round regular season in a hub environment [15]. Bowes et al. [13] further explain the gendered effects of the pandemic on women athletes from a range of sports in the UK. Longer absence from competition, in combination with less access to equipment, led to increased injury risk for elite sportswomen on their return to competition, compared to their male counterparts [13]. Thus, the inherent gender inequality in sport was exacerbated by the impact of COVID-19 on competition, training and financial security.

## Women's national basketball league 2020 season

Basketball Australia, the national sporting organization representing basketball in Australia, implemented a hub model for the entirety of its professional Women's National Basketball League (WNBL). All eight WNBL teams relocated to Queensland for the duration of a condensed six-week season (compared to the previous year's sixteen-week season) between the 12th of November and the 20th of December 2020 [17]. Games and player accommodation were spread across the cities of Cairns, Townsville and Mackay (located in the north of the State, between 1650kms and 950kms from the state capital), with athletes travelling between these three locations to play [17]. In response to the condensed season, clubs increased their player roster from ten players to twelve, and Basketball Australia implemented new structures and processes to prioritize player wellbeing, including access to: recovery centers; independent welfare and medical support; and child-minding services for players who were primary carers

of children [18]. To make the season financially viable, a league-wide pay cut of 15% was implemented for all athletes being paid above the minimum base wage of $AUD13,000 for the 2020 season [18, 19].

The WNBL's 2020 season presented a unique and unprecedented opportunity to examine the impact of the hub model on elite female athletes, including the demands of competing during a global pandemic, being away from home for the duration of a season, living with team members, coaches and support staff full-time and the physical and emotional stressors of an intense playing schedule. The hub-based season also presented an opportunity to explore the inherent strengths and limitations of the regular non-pandemic competition through the lens of the players' hub experience.

Furthermore, the impact of COVID-19 and associated restrictions on athlete wellbeing is not well documented. In their recent study, Defreese and Smith [20] found that the literature around the ways in which burnout develops in athletes is minimal, but that some athletes may develop burnout as a result of exhaustion over the course of a regular (non-hub) elite sport season, or others may develop burnout due to experiences of reduced accomplishment or devaluation. Therefore, a hub environment, comprising a condensed season where athletes are removed from their typical support structures and routines, may place athletes at greater risk of burnout. Hence it is useful to understand how perceptions of fatigue, wellbeing, performance and motivation change over the course of the season in order to gain an understanding of the impact of a hub environment during the COVID-19 pandemic on athlete burnout.

## Job demands-resources model

In order to determine the impact of the hub on athlete wellbeing, performance and burnout, it is important to identify the stressors or demands that exist for athletes in the hub environment, and what supports are most useful to help athletes meet those demands. In this study, we employ the Job Demands-Resources (JD-R) model [8, 21], as a framework to examine players' hub experiences.

Demerouti et al. [8] posited that employee wellbeing is a function of the demands of a particular job and the resources available to address or offset these demands. Excessive job demands require ongoing cognitive or emotional efforts to meet these demands, which may result in exhaustion, and psychological or physiological harm to employees [9]. Job demands may include a high volume of work (overload); complex work or boring work (underload); poor working relationships; emotionally draining work; and unclear goals or role ambiguity [22, 23]. Job resources are thought to compensate for the stress of high job demands, initiating motivational processes which lead to goal attainment and employee growth and development [9]. Such resources include training and development opportunities; autonomy over one's work; participative decision-making; clear goals; role clarity; and social relations that support and empower employees [23]. The absence of resources required to perform job requirements can result in lower levels of employee motivation, job-related cynicism and poor job performance [21].

Empirical studies show that jobs with high demands often deplete employees' physical and mental resources, resulting in exhaustion and poor health outcomes. In a study of Dutch call center workers, Bakker, Demerouti and Schaufeli [21] found that job demands were significant predictors of poor health outcomes for employees, resulting in increased employee absenteeism. However, when employees of the call center were able to access job resources, including social support and performance feedback, they were more involved and committed to their work, and were less likely to resign. van Woerkom, Bakker and Nishii [24] posited that further research is needed to explore the efficacy of different types of job resources in meeting complex

job demands, suggesting that some job resources (such as training and development or social support) may not offer commensurate utility to employees who are suffering the compounding effects of various job demands. Whilst heightened job demands are the strongest predictor of burnout amongst employees [25], some studies suggest that having in sufficient work to occupy one's time (or role underload) may also result in negative outcomes for workers [26]. Sales [27] and Shultz et al. [28] argued that workers experienced boredom and disinterest when they had capacity to undertake many more activities during their working day. A study of members of a symphony orchestra by Parasuraman and Purohit [26] found that the under-utilization of musician's skills resulted in boredom stress, which was exacerbated by long and repetitive rehearsals in which not all musicians were actively involved across the sessions.

Increased stress and decreased motivation in athletes have been linked to experiences of burnout, which in turn can lead to reduced athletic performance, and impaired physical and psychological health [29]. The JD-R model has previously been applied in an elite sport context, where elite sport has effectively become a job for many athletes and as it becomes more professional, sport becomes closer to work than play [30]. Balk et al. [31], used an extension of the JDR model, the Demand-Induced Strain Compensation (DISC) Model to gain an understanding of the interplay between job demands and resources for professional and semi-professional athletes. They found that the increasing professionalization of elite sport has led to athletes experiencing greater physical, emotional and cognitive demands [32]. The demands on elite athletes must be balanced with 'matched' physical, emotional and cognitive resources in order to promote athlete wellbeing and elite performance [31, 33, 34]. Balk et al. [32] found that physical and emotional 'detachment' from the sport are essential resources for athletes to aid their recovery from the demands of sport. It is hypothesized that the hub environment, characterized by significant periods of time away from friends and family, and limited opportunities for athletes to 'detach' or step away from the sport, will increase demands, leading to physical and emotional exhaustion and hence higher risk of burnout.

By developing an understanding of the job demands that exist in the hub and which resources are useful to counteract them, this research will provide an understanding of how the hub impacts on athletes over time. This study used qualitative data from multiple, sequential semi-structured interviews to understand WNBL player's perceptions of the hub over time, how their wellbeing and performance was affected, and whether feelings of burnout were experienced.

## Method

All 96 WNBL athletes playing in the 2020 season were invited to participate in this study by a senior staff member of Basketball Australia. Prior to recruitment, he Human Research Ethics Committee approved this study, number HEC20458, and at the beginning of each participant's first interview, oral consent was obtained. We conducted 20 semi-structured interviews with nine athletes playing in the 2020 WNBL season (interviewing approximately one in every 10 athletes). Players were asked to express their interest in participating in the study by completing an online expression of interest (EOI) form. Eleven athletes completed the EOI, who were contacted and provided with additional information about the study. Nine athletes agreed to participate in the study and their demographic characteristics are presented in Table 1 below.

To explore the impact of the hub on athlete wellbeing and performance, multiple, sequential interviews were held with each player during the 6-week season (excluding the finals). This approach was applied as it was anticipated that the athlete experience would change over the course of the hub and this would foster richer insights. Three athletes participated in three

**Table 1. Participant demographics.**

| Characteristic | Category | # Participants |
|---|---|---|
| **Level of experience in the WNBL** | Low (0–2 years) | 3 |
| | Medium (3–5 years) | 4 |
| | High (6+ Years) | 2 |
| **Country of Birth** | Australia | 7 |
| | United States of America | 2 |
| **WNBL Club** | Adelaide | 1 |
| | Bendigo | 3 |
| | Melbourne | 2 |
| | Perth | 1 |
| | Sydney | 2 |
| **Leadership Group** | Yes | 3 |
| | No | 6 |
| **Employment status during the hub (not including WNBL-related work)** | Not working | 6 |
| | Part time | 0 |
| | Dropped from full time to part time | 2 |
| | Full time | 1 |
| **Level of academic study undertaken while in the hub** | Not Studying | 4 |
| | Online Diploma | 4 |
| | Part time university | 0 |
| | Full time university | 1 |

interviews each, five athletes participated in two interviews each, and one athlete participated in one interview.

Interviews were conducted over the phone (n = 18) or via video conferencing (n = 2), depending on the athlete's preference. On average, each interview was 30 minutes duration, which, with the multiple sequential method and considering the limited time athletes had available away from their basketball duties and recovery time, allowed data saturation to be reached and no new themes emerged in the third interviews with participants. Interviewees were asked a range of questions about their experience of being an elite athlete in a hub environment and how it impacted their wellbeing, as well as what they were finding demanding in the hub, and what resources they accessed or were aware of to support them during the competition. Refer to Appendix 1 for the full interview guide.

The interviews were recorded and transcribed verbatim immediately following each interview. The transcript of each interview was returned to the interviewee for verification. Pseudonyms were used to ensure the anonymity of the athletes. The data was then imported into NVIVO 12 for analysis. The data was analyzed thematically, using the approach outlined by Braun and Clarke [35]. Thematic analysis involves the identification and analysis of patterns within qualitative data. In this study, two approaches to thematic analysis were employed. Some themes were generated inductively, with the development of themes being driven by the athlete's voice. Other themes were derived from existing theories (deductive thematic analysis), particularly those relating to job demands and job resources that were well defined in the literature and experienced by the athletes during the hub. Themes were organized hierarchically, with broad themes being broken down into more specific subthemes. Once the initial coding was completed, one third of the transcripts were independently analyzed by a second coder, using the existing coding framework. Through this process, the definitions of themes and sub-

themes were refined. Inter-coder reliability was measured by calculating the level of agreement between each coder for each theme, using Cohen's [36] Kappa value. The reliability of the coding framework was high, with an average kappa value of (0.93). Refer to Appendix 2 for the coding framework.

## Results

The results are presented in three sections. The first focuses on the longitudinal data and how the athletes' experiences of the hub changed over the course of the six-week season through a vignette of one player's experience. The second section covers the job demands experienced by athletes, and the third addresses the job resources that were available in the hub.

### Effects of the hub on athletes over time

By undertaking multiple sequential interviews with participants, we gained an understanding of how the hub life impacted participants over time, and explored whether experiences of burnout intensifies the longer the individual was situated in that environment. Due to the condensed nature of the season, not all participants were able to find time for three interviews over the six weeks. Player 4 participated in all three interviews (one each in weeks two, four and six of the truncated season) and their experience is highlighted as a vignette.

In their first interview, Player 4 was already discussing some challenges in the hub, particularly being constantly surrounded by their teammates, but overall was positive about how it was working and the steps they were taking to detach from the sport and make sure the team atmosphere was positive:

> I'd say 75% doing well. Then you're going to have to struggle, have your ups and downs, but majority of it has been good. . . We tried to keep our days off real to ourselves because we don't want. . . We know we're going to be on top of each other all the time so we really try to limit team stuff. (Player 4.1)

Just after the halfway point in week four, Player 4 experienced 'time dragging on' and talked about the difficulty of being away from home and being unable to help with family issues. This negatively impacted their wellbeing and was different to a regular season where they would be spending the majority of time at home surrounded by social support structures:

> Everything's kind of almost compacting. There's a lot of family stuff going on for me back home and yeah, it's just been a bit harder to manage, and I've had some consecutive bad days, which it's not easy, but yeah, trying to figure it all out. . . it was just a bit more tough I would say, this last week. (Player 4.2)

By the last week of the competition, Player 4 had lost motivation and wanted to go home. They experienced some challenges communicating with their coach and poor team performances which they found frustrating:

> I'm definitely over it. I don't know. It's been hard. Since last time we spoke, it's been more challenging for sure. . . just trying to get through it, to be honest at this point. (Player 4.3)

Player 4's team also did not have a full-time physiotherapist employed in the hub, so they had limited physiotherapy appointments for recovery. By the end of the season Player 4

discussed the result of this, and the negative impact the hub and lack of physiotherapy was having on her physical wellbeing:

> I think my body is just a bit worn down. The other day we had physio, but we only had a two-hour block for it. So it was girls that play top minutes [who] had the priority of the sessions. Since that was the game I played two minutes, that wasn't me. Then I was like, "I need to see the physio". My back is quite sore just from getting hit all the time. (Player 4.3)

By examining this participant's responses to the same questions at different time periods we gained insights into the athlete's mental and physical wellbeing, their outlook, and their motivation to be in the hub, which all decreased over time. This was similarly evident in other athletes who participated in two or three interviews. Themes also began to emerge in the data, and these are explored in the thematic analysis of job demands and job resources below.

## Job demands

The concept of demands refers to characteristics of the hub that participants perceived to be difficult or requiring physical, emotional or cognitive effort, likely adding to the decreased motivation over time, as noted above. The demands that emerged from the analysis and that will be unpacked in the following section are: volume of work, simultaneous overload and underload, and nature of work.

**Volume of work.** Athletes in the hub had to undertake a high volume of basketball-related work in a condensed season, which included up to four games per week. Training, active recovery, whiteboard and film sessions, as well as game preparation before each game, were also activities athletes had to engage in. In some teams, players who did not play many minutes during a game reportedly also had to complete 'top-up' fitness sessions the next day, in addition to the full team training, in order to maintain their conditioning. During a typical basketball season, players said they would usually compete in one game per week and all other sessions would be evenly spread over several days, affording more rest time for athletes and the opportunity to fit in other pursuits. Therefore, the hub created a high workload in a condensed amount of time, which led to physical and psychological overload for all participants, evidenced in the quote below:

> We have a bunch of meetings because that's the way we get better, it's because we have to have game film because we can't get on the court, and so they tried to pinch everything together because before it was like, okay, we'd have physio at this time. And then an hour and a half later we'd have a meeting and then an hour and a half later we'd have to go team recovery. So even though we didn't have training the whole day, we were going the whole day and it was just so much, it was so much. (Player 3.2)

Interestingly, despite this experience of a high volume of basketball-related work, some participants discussed experiencing underload whilst in the hub. When participants were not training, playing, recoverin or preparing for games, there were limited options available to take their mind off basketball, keep them occupied, or 'detach'. While there were break times scheduled, participants found it hard to use them to relax or wind down:

> There is time for breaks. However, I think because of the environment that we're in, I constantly feel like I'm on edge. So, I'm constantly waiting for what you have to do next or where we need to go. And I feel like we're constantly moving. (Player 8.1)

In addition to the inability to 'detach' caused by a lack of activities away from basketball, participants reported a lack of time on the basketball court to get 'shots up' and get their 'eye in'. There were limited basketball courts in the hub cities that teams could access, and they were rostered between all teams for formal training. Therefore, participants were only allowed on a court during rostered training times and were unable to access a court during their free time to have individual 'shoot-arounds' with the court to themselves. Many participants use shooting practice to relax and build confidence. Losing this crucial outlet disrupted routines, worsened the boredom and added to the emotional demands of the hub:

> You just miss out on getting their little touch shots that we would normally have. I think that's disrupting a lot of players. Players who have routines. . . been thrown in the deep end, but there's no way you can do a normal routine. So, some people really struggle with that. (Player 5.1)

**Simultaneous overload and underload.** Some participants described an interesting dichotomy of experiencing both role overload and role underload throughout their stay in the hub. Player 7.1 lamented:

> It's really frustrating because when we are not at training, I'm just sitting around twiddling my thumbs or getting into YouTube holes. But then, if we play for under 10 minutes (in a game), we have to do an extra workout session for that day. (Player 7.1)

This highlights the lack of stimulation and under-utilization of players' skills and abilities away from the court on the one hand, and the role overload experienced by the same player, having to engage with the intensity of match day and travel requirements and then being required to complete additional training to maintain their match fitness should they not receive enough game time.

This experience of competing demands was also echoed by Player 9.1 who described having 'a lot of time for breaks. . .. It's a bit too much' while also recounting the physical exhaustion she experienced from the condensed calendar:

> [Our] bodies are sorer than in a normal season when you've got seven days to recover after a game. [In the hub], you're playing two sometimes three games per week, travelling, train-ing every day, lifting every day. (Player 9.1)

Athletes were removed from their usual support structures and routines in the hub, which at home would often reportedly include low intensity shooting sessions. This was summed up by Player 9.1: 'The people that don't play much, we haven't had any time on court to shoot, which has been a bit of a struggle. I wish we had had more time'. In the hub, because of the extra physical demands of the condensed season, as well as a lack of courts available, partici-pants either experienced periods of high intensity of periods of boredom and there was a lack of of relaxed basketball opportunities which could have filled the athletes' downtime and allowed them to maintain their routine without adding physical load.

**Nature of work in the hub.** As well as the amount of work, the nature of the work partici-pants undertook in the hub was also demanding. Several characteristics of the hub added to the demands that participants reported experiencing and brought to light issues that may not have been as evident during a regular season.

All participants expressed the feeling that uncertainty and a lack of routine was inherent in the hub, as referred to above. Furthermore, schedules were often set at the last minute, with

athletes having limited advance notice of when they were training or scheduled for other team activities.

> We generally know when we're going to have a training, when we're going to have games and shoot around, but it's all of the other meetings in between. It's film sessions, or just a team meeting, or something like that will generally get dropped on us the night beforehand, so it makes it hard to plan things. (Player 6.2)

Athletes' pre-game routines were also severely disrupted. In addition to the impact of a lack of courts for training discussed above, three athletes discussed their on-court warm-up being cut from up to two hours to 20 minutes if their team was playing in the second game of the night. For these participants, this was a key part of their routine and confidence building before the game, and they found losing it cognitively demanding:

> It's that second game. . . We walked out and we had 18 minutes to do warm up, and it just was so rushed. Yeah. That was pretty challenging. Obviously, on a normal WNBL season, you get out there at the game two hours before, you just get some shots up, get some ball handling, get your mind in the game. (Player 2.1)

Staying in hotels with teammates for the duration of the season was also a significant change compared to previous seasons when participants reported that they would usually stay at home except for occasional overnight trips to away games. As a result, participants were unable to get a break, not only from their basketball work but also from their teammates. This was a significant demand that was emotionally draining and exhausting for many of the participants:

> I will say that one thing that I found difficult is, I'm the type of person who really needs to. . . I'm definitely an extrovert, but I'm also an introvert as in, I get my energy from having my own time alone and stuff. . . It felt like I didn't have any time on my own. When I finally got back to my apartment, it was time to go to sleep. And so, I had no time to decompress. (Player 3.1)

Finally, the hub environment that the athletes were working in was often stressful or physically uncomfortable, which increased the demands on the participants. Uncomfortable beds, in combination with late games that were hard to wind down from, led participants to report difficulty sleeping. This chronic sense of tiredness negatively impacted participants' emotional state and their capacity to deal with the demands of the hub, particularly towards the end of the season. Consequently, participants described feeling emotionally drained by the end of the season and a desire for the season to be over.

## Job resources

This section presents the job resources that interview participants reported were helpful to reduce job demands and improve their wellbeing and performance in the hub. An important caveat to these resources was that although they existed, many were not equally efficacious or equitably distributed across individuals or teams, and all participants reported decreased wellbeing or reduced motivation towards the end of the hub, indicating the resources were not sufficient to counteract the significant demands reported above. The resources identified by participants are based around the following four themes: recovery services; control and player agency; role clarity; and constructive social relations.

**Recovery services.** Several support services were provided to the WNBL players by Basketball Australia (BA) and the Player's Association (PA), including the physical recovery centers, ice baths and massage therapists that all participants reported being able to access. However, participants did not report accessing them equally and many would have liked to access them more. Some participants reportedly did not use them, finding it hard to fit into their busy schedule (as outlined above), or found the services provided were unsuitable. Furthermore, participants disclosed that often the ice baths were not cold enough or there was only enough ice for one team. Massages were available in the recovery centers but participants that were older and had more experience in the WNBL asserted the need for longer and more frequent massages to help them recover more effectively. Participants reported that the recovery services provided were insufficient to counter the physical demands of the condensed season:

> We went there [to the recovery center] and checked it out and it was just really basic and stuff we could do here. And I think we were told it was going to be something like plunge pools and stuff and it hasn't been that. So, we're like, "Okay, we'll just do pool recovery and get on a foam roller," kind of thing. . . In Townsville, they had ice baths, so we were using those after games. Here, they are providing one ice bath at the game. But the last game we played, the team before used all the ice, so we couldn't have one. (Player 5.1)

Clubs also provided some recovery services to their athletes, in particular physiotherapists. However, the volume and availability of physiotherapists reported by participants differed between the teams, which was likely due to the financial position and different ownership structures of the teams. Some participants discussed that their club had a physiotherapist travelling with the team for the duration of the hub to prevent and manage injuries and recovery, and participants from one club had their own massage therapists available after every game, while other clubs did not have a dedicated physiotherapist travelling with them so they had to outsource physios from local clinics for limited periods each day. This was an inequitable distribution of resources that facilitated greater recovery opportunities for some players/teams who could access physiotherapy whenever they needed rather than at set times, allowing them to cope better physically with the demands of the season. Furthermore, in the hub, participants discussed being more aware of the inequity between teams because it was more obvious when teams were travelling together and could observe these resources in action:

> When we played [team] the other day, we got back, some of us hopped in the pool to do recovery, and they've got the physios and massage people all there. So, there was probably four of them on a bed getting massages. And I was just like, "God." And like they would have done that throughout the whole team. Whoever needed it, it wouldn't just be based on how many minutes you're playing and things like that. It's just we did not have access to any of that. (Player 2.3)

**Control and player agency.** A sense of autonomy or control over the work was important to participants, particularly in response to the uncertainty and lack of routine reported above as a demand. Autonomy is also an important resource that can help reduce experiences of burnout [29]. Interview participants with less experience in the WNBL (0–2 years) enjoyed the control they had over their downtime. They reported sufficient, sometimes too much, rest and downtime away from basketball and enjoyed the fact that were not required to tell the coaches what they did or where they were going when they did not have a team activity scheduled. Unfortunately, as discussed previously, there were limited activities to do in the hub that did

not pertain to basketball, and even though they were physically resting they were not cognitively or emotionally resting, or 'detaching', as they were still in the hub environment and surrounded by teammates, leading to feelings of underload. Players with more experience in the WNBL (three or more years) talked more about having control over training or their basketball in general as a resource. Participants reportedly enjoyed being able to get on court as a team, and individually, without coaches or staff to work on areas of weakness alone, although as noted this time was often difficult to access in the hub:

> The first time when I got a session to go shoot [on my own] in Cairns, outside of practice and stuff, I was shooting up, put my headphones on and I just started crying as I was shooting, but it was a good cry. It was like. . . It was a release that I had just been holding onto for so long. (Player 3.2)

**Constructive social relations.**   Good working relationships, with teammates, family and friends, mentors, and between teams, were a significant resource for all interview participants to counteract the demands of the hub. Participants spent the most amount of time with their teammates, who were going through the same demands and experiences. This shared struggle throughout the duration of the hub brought the interview participants closer to their teammates and in some teams helped them to form lifelong bonds, as described by Player Nine who only participated in one interview in the final week of the season:

> These are girls that I think I'm going to be friends with forever now. So, it's been so good. The things we've been doing together, outside of basketball, like how I said we were going to Magnetic Island, none of that has been a forced team bonding thing. We all want to go hang out together all the time. (Player 9.1)

As well as friendships within the team, informal mentoring reportedly took place to help less experienced players navigate the challenges of playing elite sport, and the new hub environment. Informal mentoring was easier to access in the 2020 WNBL season compared to previous seasons because, due to their constant proximity, younger players were able to see every part of the more experienced players' routine and learn how they operate when under pressure, developing their skills on and off the basketball court.

Participants who had strong family or external support also considered relationships external to basketball as a resource. Even though they were isolated in a hub, having family or friends away from basketball that they could talk to on the phone, or come to Queensland and watch the games, was helpful in dealing with the cognitive demand of limited time away from basketball:

> A couple of the girls have had their partners come out, which has been great. And then one of our team members is actually, she's from Cairns, so she had a lot of her family at the games. And then yeah, a couple of other girls their husbands are coming up in the next couple of days. So that'll be really good for them. A couple of players, it's nice to have supporters in the crowd and they'll probably be super loud. But yeah, I think it will just be really good for them. They'll probably be a lot happier with their husbands up here. (Player 7.2)

Finally, relationships between players on different teams were also a key resource. Unlike past seasons, teams were living and working at the same hotel and basketball courts for extended periods of time. Particularly for older or more experienced participants who had played on other teams, this meant they had an opportunity to spend time with friends who

were not on their team, but still understood the mutual challenges of the hub, a unique resource in the 2020 season.

## Discussion

This research explores the experience of elite women athletes living and working in a COVID-19 safe hub in Queensland, Australia for the duration of the 2020 WNBL season. The research contributes to the understanding of the management of elite sport competitions in a hub environment during a pandemic by exploring how the hub impacted on athlete wellbeing and performance. The study also extends Bakker and Demerouti's JD-R model (2014) into a new and unusual context. The application of the JD-R model in this qualitative study has used the athlete's voice to understand the effects of the hub on athletes over time and enabled the emergence of key themes that participants found challenging or demanding in the hub, as well as the supports or resources that were provided to athletes across multiple organizations (or different clubs) to counter those demands. It was particularly useful to develop an understanding that in the WNBL hub the resources were not sufficient to counter the demands for athletes, resulting in a higher risk of burnout, characterized by reduced motivation as well as physical and emotional exhaustion among participants. This finding adds to van Woerkom et al.'s [24] research that certain job resources may not offer commensurate utility to workers experiencing compounding job demands. This knowledge led to new insights concerning what could be improved in future elite sport league management, in order to promote athlete wellbeing and performance, discussed below.

The hub was an intense and condensed version of a normal WNBL season, and as such existing demands were more obvious or exacerbated for athletes and emerged more clearly in the data. Overload was heightened in the hub compared to previous seasons because the games were closer together and teams had to fit training and other work or study around them, which athletes discussed leading to both physical and emotional fatigue that compounded over the course of the hub. Physical recovery services were provided to deal with overload, including recovery centers, ice baths, and physiotherapy services, but they were inequitably distributed or not sufficient for athlete needs, and did not address cognitive or emotional demands. The inequity of resource provision was in part due to the different ownership structures of the clubs involved in the WNBL, and the lack of media exposure and funding that women sports are given, which also exists in a regular WNBL season [13, 37]. This was more obvious to interview participants in the hub than in a regular season because they were living and working in close proximity to other teams that had different levels of support services. By applying the JD-R theory across multiple organizations, in this study we were able to determine that although resources were provided to athletes, the relativity of the distribution of those resources actually became a demand for athletes that would not exist in a non-hub environment. In the future it would be of benefit to athletes, both physically and cognitively, to have equity of resources across all teams.

As well as living in close proximity to other teams, athletes were living with and constantly surrounded by their own teammates and trying to fit in basketball-related work and activities. Therefore, there was less downtime available for athletes to stop thinking about basketball or work on something other than basketball, in order to 'detach' [32]. This lack of downtime was also cognitively demanding because they are unable to get a mental break from their basketball work, resulting in a high volume of work and work pressure [9]. The interplay between this experience and the underload that emerged was a unique characteristic of the hub, due to the combination of disrupted routines and a condensed season and hub conditions that restricted player movement and engagement with family, friends and other support networks outside the immediate competition. In the hub, athletes used new social structures as resources to

counteract underload, including team bonding and day trips, informal mentoring and socializing with athletes on other teams. This is consistent with the literature that states social factors influence the motivation and stress factors that can lead to burnout [38]. Therefore, in future seasons, a greater focus on inter-club socializing, as well as a more formalized mentoring program within clubs could be rewarding and help to develop athletes as individuals while avoiding additional physical load, whether a hub model is implemented or not [39]. Whilst Fisher [40] posed that stress may result from role underload or role overload, few studies explore how the contemporaneous experience of role overload and underload impacts on the experience of work.

Furthermore, while uncertainty was inherent in the changing landscape of a global pandemic, more could have been done to ask participants how much time and court space they needed for their pre- and post-game routines. This consultation could then have been used to give more control to athletes in building a more consistent and appropriate schedule (weekly and daily), as well as increased court time for low intensity shoot arounds in order to combat both overload and underload, acknowledging that with limited courts this will always be a challenge. Taking athlete perspectives into account in this way, and giving them greater control and agency over their work, is also understood to be an essential cognitive resource which can reduce burnout risk [38].

Finally, it was clear that towards the end of the season, compared to the first two weeks, participants were physically and emotionally exhausted. This resulted in reduced motivation and feelings of wanting to go home rather than continue competing. Elite sport operates in a complex organizational environment and is an inherently stressful enterprise [41]. As demonstrated by the sequential nature of the data and the application of the JD-R model, in the hub model the inherent elite sport stress was exacerbated and compounded over time. The resources provided in the hub were often inequitably distributed and insufficient to counteract the stressors or demands, resulting in an increase in athlete stress, and a decrease in motivation and performance. As noted above, increased stress, fatigue and decreased motivation in athletes is likely to lead to experiences of burnout [29, 42]. Athletes in a hub environment for an extended period of time are therefore at an increased risk of burnout, negatively impacting their wellbeing and performance. It is likely that this would also be true for athletes in other sports who experience living and working in a hub environment.

## Conclusion

The use of the JD-R model in a qualitative study was successful in determining that there was a gap between resources provided to participants and the demands unique to the WNBL hub model, in addition to the demands that already existed for athletes in the WNBL. To the best of our knowledge these demands, the efficacy of the resources provided to respond to the demands, and athletes' experiences of them, have not before been reported in the literature at the time of writing. The intensity of the hub model for athletes also likely magnified issues present in all WNBL seasons. Therefore, by understanding the experience of the athletes in this new and unusual setting, this study allows us to understand the impact of the hub on athlete wellbeing and performance in both hub and other WNBL season models. These results are helpful for the management of elite sport leagues and teams, in particular when considering new settings where athletes' routines are disrupted, and they are taken away from their home base and support networks for extended periods of time.

## Supporting information

**S1 Table. Interview guide.** Main concepts and example questions for each interview. (DOCX)

**S2 Table. Coding structure.** Nodes and examples of sub-nodes that were used to code the qualitative data.
(DOCX)

## Author Contributions

**Conceptualization:** Samantha Marshall, Nicola McNeil, Emma Louise Seal, Matthew Nicholson.

**Data curation:** Samantha Marshall.

**Formal analysis:** Samantha Marshall.

**Investigation:** Samantha Marshall.

**Methodology:** Samantha Marshall, Nicola McNeil, Emma Louise Seal, Matthew Nicholson.

**Project administration:** Samantha Marshall.

**Supervision:** Nicola McNeil, Emma Louise Seal, Matthew Nicholson.

**Writing – original draft:** Samantha Marshall.

**Writing – review & editing:** Samantha Marshall, Nicola McNeil, Emma Louise Seal, Matthew Nicholson.

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
