## [Decision Letter · Decision Letter 0]

22 Apr 2022

PONE-D-21-40132Elite sport hubs during COVID-19: The job demands and resources that exist for athletesPLOS ONE

Dear Dr. Marshall,

Thank you for submitting your manuscript to PLOS ONE. After careful consideration, we feel that it has merit but does not fully meet PLOS ONE’s publication criteria as it currently stands. Therefore, we invite you to submit a revised version of the manuscript that addresses the points raised during the review process.

Please address the comments by the reviewers addressing clarity of writing and references.

We look forward to receiving your revised manuscript.

Kind regards,

Andrew Philip Lavender, PhD

Academic Editor

PLOS ONE

Journal Requirements:

This research was supported by a La Trobe University and Basketball Australia

This study was supported by a PhD Industry Scholarship with Basketball Australia. The funders had no role in study design, data collection and analysis, decision to publish, or preparation of the manuscript.

Reviewers' comments:

Reviewer's Responses to Questions

**Comments to the Author**

1. Is the manuscript technically sound, and do the data support the conclusions?

Reviewer #1: Yes

Reviewer #2: Yes

2. Has the statistical analysis been performed appropriately and rigorously? 

Reviewer #1: Yes

Reviewer #2: Yes

3. Have the authors made all data underlying the findings in their manuscript fully available?

Reviewer #1: Yes

Reviewer #2: Yes

4. Is the manuscript presented in an intelligible fashion and written in standard English?

Reviewer #1: Yes

Reviewer #2: No

5. Review Comments to the Author

Reviewer #1: The aim of the study was to qualitatively investigate the impacts of a hub model on athletes over time, and what job demands and resources existed for athletes based on the Job Demands-Resources Model. Overall this is a very interesting, well presented manuscript. It address a contemporary issue in elite sport, and more importantly, focuses on the lived experience of female athletes - an area seldom explored. The following comments are made in an effort to help the author further improve the quality of their manuscript.

Introduction is well written, and provides a complete background of relevant literature to setup for the results and discussion.

Page 4 Line 68 - suggest inclusion of a reference here

Page 5 Line 94 - its not immediately clear what is/ are "burnout pathways" - suggest to provide some explanation of what this is, or alternatively improve clarity around terminology here (e.g., ... direct and indirect causes of burnout in athletes is minimal")

Page 7 Line 146 Is there a reason the DISC model was not used in your study? Particularly as it was stated that it has been used in both elite and semi elite athletes

Suggest to provide some further clarity around methodology.

Page 9 Line 185 Was the maximum allotted time you had (30min) sufficient for data to reach a (theoretical) saturation point?

Similar to the above query, did any new themes continue to emerge?

Page 8 Line 192 Suggest to consider providing some exemplar questions used in the interviewing to improve transparency

Results section is well laid out and use of subheadings clearly guide the reader

Page 23 Line534-534 Suggest this sentence needs to be more clearly linked to data/ results section

Reviewer #2: Thanks for the opportunity to read such an interesting paper.

The only revision am suggesting realties to the lit review. there are many NEW papers that can be added, and most importantly, to discuss in relation to the paper's finds. Those include:

Samuel, R. D., Tenenbaum, G., & Galily, Y. (2020). The 2020 coronavirus pandemic as a change-event in sport performers’ careers: conceptual and applied practice considerations. Frontiers in Psychology, 2522.‏

Pons, J., Ramis, Y., Alcaraz, S., Jordana, A., Borrueco, M., & Torregrossa, M. (2020). Where did all the sport go? negative impact of COVID-19 lockdown on life-spheres and mental health of spanish young athletes. Frontiers in Psychology, 11, 3498.‏

Fiorilli, G., Grazioli, E., Buonsenso, A., Di Martino, G., Despina, T., Calcagno, G., & Di Cagno, A. (2021). A national COVID-19 quarantine survey and its impact on the Italian sports community: Implications and recommendations. Plos one, 16(3), e0248345.‏

Washif, J. A., Farooq, A., Krug, I., Pyne, D. B., Verhagen, E., Taylor, L., ... & Chamari, K. (2022). Training during the COVID-19 lockdown: Knowledge, beliefs, and practices of 12,526 athletes from 142 countries and six continents. Sports Medicine, 52(4), 933-948.‏

6. PLOS authors have the option to publish the peer review history of their article (what does this mean?). If published, this will include your full peer review and any attached files.

Reviewer #1: No

Reviewer #2: **Yes: **Yair Galily

---

## [Author Response · Author response to Decision Letter 0]

25 May 2022

Thank you for these helpful comments and suggestions to improve our work. Similar to the Response to Reviewers, please see below each comment, and the action taken in response to that comment. 

1. Page 4 Line 68 - suggest inclusion of a reference here Page 4 Line 75 now includes the reference: 

Bowes et al., 2020 (13) has been included. Text now reads: ‘…compared to their male counterparts (13).’

2. Page 5 Line 94 - its not immediately clear what is/ are "burnout pathways" - suggest to provide some explanation of what this is, or alternatively improve clarity around terminology here (e.g., ... direct and indirect causes of burnout in athletes is minimal") 

Page 5 Line 101 has been changed to: ‘…the literature around the ways in which burnout develops in athletes is minimal, but that some athletes may experience burnout as a result of …’

3. Page 7 Line 146 Is there a reason the DISC model was not used in your study? Particularly as it was stated that it has been used in both elite and semi elite athletes 

Whilst the DISC model, which focuses on ‘matching’ job demands with job resources to mitigate the demands, has been used in the sport context previously, in this study the authors made the choice to use the original JDR model instead. The goal of this preliminary study was to gain a broad understanding of the demands and resources that WNBL athletes experienced in the hub, not to match them. Thus, the DISC model was not appropriate as we did not seek to match demands and resources. 

4. Suggest to provide some further clarity around methodology.

a. Page 9 Line 185 Was the maximum allotted time you had (30min) sufficient for data to reach a (theoretical) saturation point?

b. Similar to the above query, did any new themes continue to emerge?

c. Page 8 Line 192 Suggest to consider providing some exemplar questions used in the interviewing to improve transparency 

Changed to, on Page 10, Line 192: “On average, each interview was 30 minutes duration, which, with the multiple sequential method and considering the limited time athletes had available away from their basketball duties and recovery time, allowed data saturation to be reached and no new themes emerged in the third interviews with participants.”

The full interview guide is provided in the appendices, Page 31, line 700, to maximize transparency of our methods. 

5. Page 23 Line534-534 Suggest this sentence needs to be more clearly linked to data/ results section.

Changed to, at Page 24, Line 540: Taking athlete perspectives into account in this way, and giving them greater control and agency over their work, is also understood to be an essential cognitive resource…”

Athlete control and agency is a key section of the results. 

6. Samuel, R. D., Tenenbaum, G., & Galily, Y. (2020). The 2020 coronavirus pandemic as a change-event in sport performers’ careers: conceptual and applied practice considerations. Frontiers in Psychology, 2522.‏ 

Added into Page 3, Line 40: Indeed, the pandemic has been conceptualized as a ‘longitudinal, multifaceted, unpredicted, non-controlled change event’ which impacted athletic careers and had varying impacts on the mental health of athletes (5, 6).

7. Pons, J., Ramis, Y., Alcaraz, S., Jordana, A., Borrueco, M., & Torregrossa, M. (2020). Where did all the sport go? negative impact of COVID-19 lockdown on life-spheres and mental health of spanish young athletes. Frontiers in Psychology, 11, 3498.‏ 

Added into Page 3, Line 40: Indeed, the pandemic has been conceptualized as a ‘longitudinal, multifaceted, unpredicted, non-controlled change event’ which impacted athletic careers and had varying impacts on the mental health of athletes (5, 6). 

8. Fiorilli, G., Grazioli, E., Buonsenso, A., Di Martino, G., Despina, T., Calcagno, G., & Di Cagno, A. (2021). A national COVID-19 quarantine survey and its impact on the Italian sports community: Implications and recommendations. Plos one, 16(3), e0248345.‏ 

Added at Page 3, Line 63: Psychologically the pandemic increased demands on athletes, both elite and amateur, and women athletes were found to be more stressed and distressed than male athletes (14). 

9. Washif, J. A., Farooq, A., Krug, I., Pyne, D. B., Verhagen, E., Taylor, L., ... & Chamari, K. (2022). Training during the COVID-19 lockdown: Knowledge, beliefs, and practices of 12,526 athletes from 142 countries and six continents. Sports Medicine, 52(4), 933-948.‏ 

Added in at Page 3, Line 54: The Covid-19 pandemic impacted athletes’ training and competition, affecting their fitness and skill maintenance and development, as well as their mental health (10).

---

## [Editor Report · Decision Letter 1]

30 May 2022

Elite sport hubs during COVID-19: The job demands and resources that exist for athletes

PONE-D-21-40132R1

Dear Dr. Marshall,

We’re pleased to inform you that your manuscript has been judged scientifically suitable for publication and will be formally accepted for publication once it meets all outstanding technical requirements.

Kind regards,

Andrew Philip Lavender, PhD

Academic Editor

PLOS ONE

---

## [Editor Report · Acceptance letter]

27 Jun 2022

PONE-D-21-40132R1 

Elite sport hubs during COVID-19: The job demands and resources that exist for athletes  

Dear Dr. Marshall:

I'm pleased to inform you that your manuscript has been deemed suitable for publication in PLOS ONE. Congratulations! Your manuscript is now with our production department. 

Kind regards, 

on behalf of

Dr. Andrew Philip Lavender 

Academic Editor

PLOS ONE